# *HUMANISE*: Language-conditioned Human Motion Generation in 3D Scenes

**Zan Wang**[1,2]**, Yixin Chen**[2]**, Tengyu Liu**[2]
**Yixin Zhu**[3✉]**, Wei Liang**[1,4✉]**, Siyuan Huang**[2✉]
✉ indicates corresponding authors
[1] School of Computer Science & Technology, Beijing Institute of Technology
[2] Beijing Institute for General Artificial Intelligence (BIGAI)
[3] Institute for Artificial Intelligence, Peking University
[4] Yangtze Delta Region Academy of Beijing Institute of Technology, Jiaxing

https://silverster98.github.io/HUMANISE/

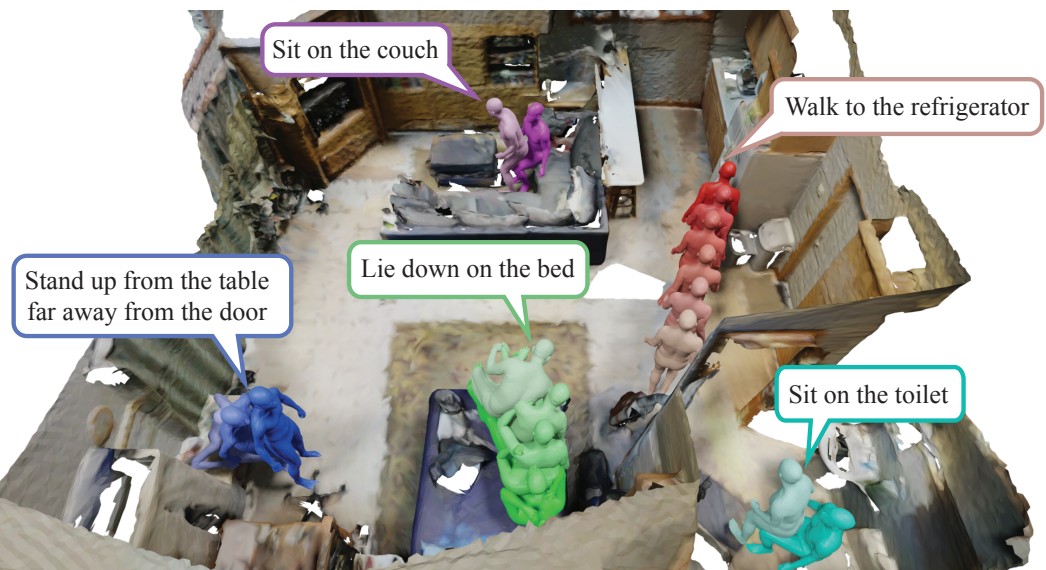

Figure 1: **Examples from our proposed Human-Scene Interaction (HSI) dataset—*HUMANISE*.** It contains 19.6k high-quality motion sequences in 643 indoor scenes. Each motion segment contains rich semantic information about the ground-truth action type and objects being interacted with, specified by language descriptions.

## Abstract

Learning to generate diverse scene-aware and goal-oriented human motions in 3D scenes remains challenging due to the mediocre characteristics of the existing datasets on Human-Scene Interaction (HSI); they only have limited scale/quality and lack semantics. To fill in the gap, we propose a large-scale and semantic-rich synthetic HSI dataset, denoted as *HUMANISE*, by aligning the captured human motion sequences with various 3D indoor scenes. We automatically annotate the aligned motions with language descriptions that depict the action and the unique interacting objects in the scene; *e.g.*, sit on the armchair near the desk. *HUMANISE* thus enables a new generation task, *language-conditioned human motion generation in 3D scenes*. The proposed task is challenging as it requires joint modeling of the 3D scene, human motion, and natural language. To tackle this task, we present a novel scene-and-language conditioned generative model that can produce 3D human motions of the desirable action interacting with the specified objects. Our experiments demonstrate that our model generates diverse and semantically consistent human motions in 3D scenes.

36th Conference on Neural Information Processing Systems (NeurIPS 2022).

# 1 Introduction

Imagine instructing an agent (*e.g.*, a virtual human or humanoid robot) to "sit on the armchair near the desk." Upon accepting the instruction, the agent needs to comprehend its semantic meaning (*i.e.*, *sit*) and understand the surrounding environment (*i.e.*, *the armchair near the desk*) to generate desired interactions with the 3D scene. Despite recent progress in human motion synthesis, generating 3D human motion conditioned on the instructions with proper scene context is still challenging.

Specifically, existing works generate plausible human poses [Zhang et al., 2020b;a] or motions [Starke et al., 2019; Cao et al., 2020; Wang et al., 2021b; Hassan et al., 2021a; Yi et al., 2022] by learning from existing HSI datasets (*e.g.*, PROX [Hassan et al., 2019], GTA-IM [Cao et al., 2020]). However, these datasets possess two fundamental issues that prevent existing methods from learning instruction-aware and generalizable 3D human motions.

- **Limited scale and quality**. PROX [Hassan et al., 2019] only contains 12 indoor scenes with jittering human motions. The synthetic GTA-IM [Cao et al., 2020] only has 49 scenes and lacks 3D human shapes and diverse movement styles.

- **Absence of scene and interaction semantics**. PROX [Hassan et al., 2019] and GTA-IM [Cao et al., 2020] both suffer from incomplete 3D semantic segmentation annotations. Particularly, the lack of human action labels/descriptions hinders its ability to generate instruction-aware human motions.

To tackles the above issues, we propose a large-scale and semantic-rich synthetic HSI dataset, *HUMANISE* (see Fig. 1), by aligning the captured human motion sequences [Mahmood et al., 2019] with the scanned indoor scenes [Dai et al., 2017]. Specifically, for an action-specific motion sequence (*e.g.*, *sit*), we first sample an interacting target object (*e.g.*, *armchair*) and the possible interacting positions on the object surface. Next, we sample the valid translation and orientation parameters by employing the *collision* and *contact* constraints, such that the interactions between the transformed agent and the scene are physically plausible and visually natural. Finally, we automatically annotate the synthesized motion sequences with template-based language descriptions; for instance, *sit on the armchair near the desk*. To specify the interacting objects, we adopt the object referential utterances in Sr3D [Achlioptas et al., 2020]. In total, *HUMANISE* contains 19.6k high-quality motion sequences in 643 indoor scenes. Each motion segment has rich semantics about the action type and the corresponding interacting objects, specified by the language description.

Based on *HUMANISE*, we propose a new task: *language-conditioned human motion generation in 3D scenes*. It requires the model to generate diverse and physically plausible human motions with designated action types and interacting 3D objects specified by the language descriptions. We argue that this task is more challenging than previous motion generation tasks in the following aspects: (i) The motion generation is conditioned on the multi-modal information including both the 3D scene and the language description. (ii) The generated human motions should perform the correct action and be precisely grounded near the target location according to the language descriptions. (iii) The generated human motions should be realistic and physically plausible within the 3D scenes. Therefore, an ideal model should equip with capabilities of *multi-modal understanding*, *language grounding*, and *physically plausible generation*.

Computationally, we leverage the conditional variational auto-encoder (cVAE) [Sohn et al., 2015] framework to generate human motions conditioned on the given scene and language description. The scene and language conditions are learned with separate encoders and fused with the self-attention mechanism to generate conditional embedding. The input motion is encoded with a recurrent module and decoded with a transformer decoder to generate human motions. We further design two auxiliary losses for object grounding and action-specific generation. The qualitative and quantitative results show that our model generates diverse and semantically consistent human motions in 3D scenes; it outperforms the baselines on various evaluation metrics. By evaluating on downstream tasks, further experiments demonstrate that *HUMANISE* alleviates the limits of existing HSI datasets.

This paper makes three primary contributions: (i) We propose a large-scale and semantic-rich synthetic HSI dataset, *HUMANISE*, that contains human motions aligned with 3D scenes and corresponding language descriptions. (ii) We introduce a new task of *language-conditioned human motion generation in 3D scenes* that requires a holistic and joint understanding of scenes, human motions, and language. (iii) We develop a generative model that produces diverse and semantically consistent human motions conditioned on the 3D scene and language description.

## 2 Related work

**Conditional human motion synthesis** Human motion synthesis can be conditioned on past motion [Li et al., 2018; Barsoum et al., 2018; Yuan and Kitani, 2020; Cao et al., 2020; Xie et al., 2021], 3D scenes [Starke et al., 2019; Wang et al., 2021a;b; Hassan et al., 2021a], objects [Chao et al., 2019; Taheri et al., 2020], action labels [Guo et al., 2020; Petrovich et al., 2021], and language descriptions [Ahuja and Morency, 2019; Ghosh et al., 2021; Lin and Amer, 2018]. Given a furnished 3D indoor scene, algorithms [Zhang et al., 2020b;a; Hassan et al., 2021b;a; Wang et al., 2021a;b] can generate physically plausible single-frame human pose or long-term human motions. However, these methods either generate random/uncontrollable human motions or require target positions due to the lack of semantics constraints. Provided textual descriptions, existing motion generation algorithms [Lin and Amer, 2018; Ahuja and Morency, 2019; Ghosh et al., 2021] focus only on motions alone and neglect the scene context. In this work, we incorporate the context of both scenes and language descriptions, generating human motion sequences consistent with the specified action and location.

**Human-Scene Interaction (HSI) datasets** Data captured in both the physical and the virtual worlds have been adopted to study HSI related topics [Wang et al., 2019; Chen et al., 2019; Liang et al., 2019; Jia et al., 2020; Wang et al., 2020; Huang et al., 2022]. PiGraphs [Savva et al., 2016], captured with RGB-D sensors, suffers from inaccurately reconstructed scenes, noisy human poses, and a low frame rate. Even with new pipelines to reconstruct human motions from monocular RGB-D sequences, PROX's pose fitting results [Hassan et al., 2019] are still visibly worse compared to high-quality, large-scale MoCap data (*e.g.*, AMASS [Mahmood et al., 2019]). GTA-IM [Cao et al., 2020] exploits the game engine to collect a synthetic dataset of human skeletons, but with limited HSIs and scene diversities. In comparison, our *HUMANISE* dataset synthesizes large-scale HSIs by aligning high-quality motions in AMASS [Mahmood et al., 2019] with real-world 3D scenes in ScanNet [Dai et al., 2017], resulting in high-quality and realistic human-scene interactions with high diversity (*i.e.*, 643 scenes, 19.6$k$ motions) and rich semantics provided by language descriptions.

**Language grounding in 3D scenes** Grounding 3D objects w.r.t. the language input has been explored in 3D reference [Achlioptas et al., 2020; Chen et al., 2021; Hong et al., 2021; Li et al., 2022; Thomason et al., 2022], 3D question answering (QA) [Das et al., 2018; Ye et al., 2021; Azuma et al., 2021], and embodied navigation [Anderson et al., 2018; Gordon et al., 2019; Gan et al., 2020]. To date, locating the interacting objects in 3D scenes with referential descriptions is still challenging [Roh et al., 2022; Chen et al., 2020; 2021] due to the noisy scanned scenes, diverse expressions, and complex spatial relation reasoning. To better learn conditional embedding that facilitates locating and grounding according to the provided language descriptions with proper scene context, we introduce a multi-model conditional module that learns a joint embedding from textual descriptions and 3D point clouds. We further enhance the model's spatial grounding and action grounding capabilities by introducing two auxiliary tasks of locating interacting 3D objects and action comprehension.

## 3 The *HUMANISE* dataset

A large-scale and high-quality dataset with rich semantic information is in need to facilitate the research in human motion synthesis and HSI. In this work, we propose a synthetic dataset by leveraging the existing datasets in human motion (*i.e.*, AMASS [Mahmood et al., 2019]), 3D indoor scene (*i.e.*, ScanNet [Dai et al., 2017]), and additional annotations of these datasets (*i.e.*, motion action label in BABEL [Punnakkal et al., 2021] and 3D referential descriptions in Sr3D [Achlioptas et al., 2020]). Our novel automatic synthesis pipeline avoids expensive hardware and time-consuming labor for realistic capture. Fig. 2 shows more samples in *HUMANISE*. Of note, our synthesis pipeline can generalize to other 3D scene datasets and actions; please refer to Appendix C for more details.

Below, we start by describing how to align the motion with the 3D scene in Sec. 3.1, followed by automatic language description generation in Sec. 3.2. Sec. 3.3 provides dataset statistics.

### 3.1 Motion alignment

The idea of synthesizing *HUMANISE* is straightforward: Automatically aligning captured human motions with 3D indoor scenes. Given a human motion sequence and its action label, we first sample an interacting target object by the predefined category labels and possible interacting positions on the object surface in the 3D scene. Next, we sample the valid global translation $\mathcal{T}$ and rotation $\mathcal{R}$ parameters by employing the *collision* and *contact* constraints; we align the human motion with the 3D scene through a transformation with $\mathcal{T}$ and $\mathcal{R}$, while all the local poses remain unchanged.

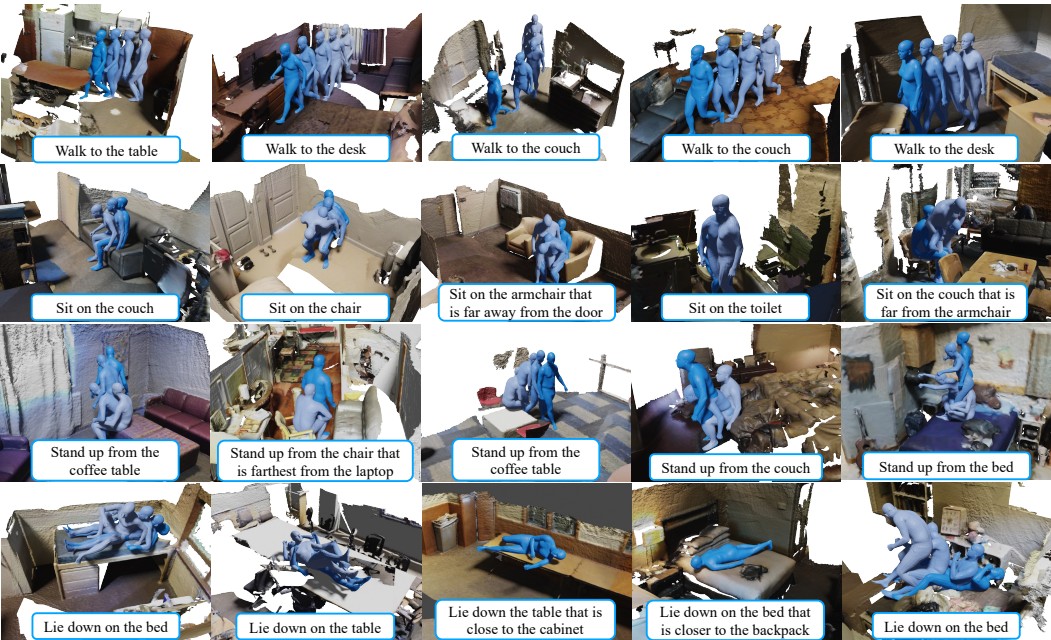

Figure 2: **Samples from the *HUMANISE* dataset**. *HUMANISE* includes representative interactive actions (*e.g.*, sit, stand-up, lie-down, walk) in diverse indoor scenes (*e.g.*, office, bedroom, living room, dining room).

**Collision constraint** We treat the point cloud of the 3D scene as a 3-dimensional $k$-d tree [Bentley, 1975]. Collision avoidance of human motions is equivalent to finding the closest point to the queried human vertex in the scene by searching through the $k$-d tree. Collision happens as the distance between the human vertex and the closest point in the scene is less than a threshold. A valid motion should avoid collisions with the objects or walls for all the poses along the trajectory.

**Contact constraint** We further decompose this constraint into the foot-ground support constraint and the proximity between the human and the interacting object. The support constraint is a distance threshold between the foot and the ground plane. The human-object proximity differs among actions; we design the following action-specific proximity constraints for realistic interaction:

- **Sit** We sample a sittable object (*e.g.*, a chair) as interacting objects and uniformly sample 3D points on the object surface as potential contact points. We assume the final pose of the sitting motion should contact the sittable object on the hip area, such that the distances between the hip vertices and the contact point should be less than a threshold.
- **Stand-up** Since the stand-up motion interacts with the sittable object at the starting frame, we enforce the similar constraints as in *sit* on the starting pose instead of the final pose.
- **Walk** The language description in *HUMANISE* provides instructions to walk to a target location, usually to an object; *e.g.*, *walk to the refrigerator*. We sample dense positions near the specified object on the floor as the potential targets for the final pose of walking motion. All the body poses along the walking trajectory should satisfy the feet-ground support and avoid collisions.
- **Lie-down** We select objects with a large flat surface (*e.g.*, bed) as the interacting objects. The translation $\mathcal{T}$ and rotation $\mathcal{R}$ are sampled such that the final pose of the lie-down motion is in contact with the object's top surface. Apart from the non-collision and contact constraint, we require the human body to lie within the object area.

**Diverse affordance** We sample motions representing diverse 3D affordance with the motion alignment algorithm, *e.g.*, *stand up from the coffee table*, *lie down on the office table*, and *sit on the toilet*. *HUMANISE* contains these Human-Object Interactions (HOIs) that are not frequently captured in daily activity or previous HSI datasets. They encode meaningful and versatile object/scene affordances, which introduce both potentials and challenges to the model building.

### 3.2 Language description generation

The motion descriptions in *HUMANISE* depict the action type and the objects/locations being interacted with. To automatically generate these language descriptions, we design a compositional

template following Sr3D [Achlioptas et al., 2020]:

$$< action >< target\text{-}class > [< spatial\text{-}relation >< anchor\text{-}class(es) >].$$

For instance, sit on the armchair near the desk. The template contains four placeholders, of which *spatial-relation* and *anchor* are optional placeholders. The *action* is taken from the motion labels in BABEL [Punnakkal et al., 2021], whereas the *target* represents the interacting object. To uniquely refer to a target object, a *spatial-relation* is defined between the *target* and a surrounding object (*anchor*). We utilize the referential utterances in Sr3D [Achlioptas et al., 2020], which defines five types of spatial relations: horizontal proximity, vertical proximity, between, allocentric, and support.

### 3.3 Dataset statistics

*HUMANISE* includes 19.6k human motion sequences in 643 3D scenes. The four most representative actions are sit (5578), stand up (3463), lie down (2343), and walk (8264). Tab. 1 compares *HUMANISE* with existing HSI datasets; *HUMANISE* shows advantages in scene variety, clip number, frame number, and human pose quality. Of note, *HUMANISE* is the only dataset with rich semantic information of HSIs. Fig. 2 shows more examples from *HUMANISE*. Please refer to Appendices A and B for more statistics and quantitative comparison between *HUMANISE* and PROX, respectively.

Table 1: **Comparison between *HUMANISE* and existing HSI datasets.**

| Dataset | #Scenes | #Clips | #Frames | Human Representation | Pose Jittering | Semantics |
|---|---|---|---|---|---|---|
| PiGraph [Savva et al., 2016] | 30 | 63 | 0.1M | Skeleton | ✓ | ✗ |
| PROX-Q [Hassan et al., 2019] | 12 | 60 | 0.1M | Shape | ✓ | ✗ |
| GTA-IM [Cao et al., 2020] | 49 | 119 | 1.0M | Skeleton | ✗ | ✗ |
| *HUMANISE* | 643 | 19.6k | 1.2M | Shape | ✗ | ✓ |

## 4 Language-conditioned human motion generation in 3D scenes

### 4.1 Problem definition and notations

Our goal is to generate a human motion sequence that is both semantically consistent with the language description and physically plausible in interacting with the scene. We denote the given scene as $S \in \mathbb{R}^{N \times 6}$, representing an RGB-colored point cloud of $N$ points. The language description is a tokenized word sequence of length $D$, denoted as $L_{1:D} = [w_1, \cdots, w_D]$. We represent the human pose and shape in the motion sequence using the SMPL-X [Pavlakos et al., 2019] body model. Specifically, the SMPL-X body mesh $\mathcal{M} \in \mathbb{R}^{10475 \times 3}$ is parameterized by $\mathcal{M} = F(t, r, \beta, p)$, where $t \in \mathbb{R}^3$ is the global translation, $r \in \mathbb{R}^6$ a continuous representation [Zhou et al., 2019] of the global orientation, $\beta \in \mathbb{R}^{10}$ the body shape parameters, $p \in \mathbb{R}^{J \times 3}$ $J$ joint rotations in axis-angle, and $F$ a differentiable linear blend skinning function. To model the effect of body shape in motion generation, we treat $\beta$ as an additional condition and denote the parameters of interest as $\Theta = \{t, r, p\}$.

### 4.2 Method

We propose a novel generative model to generate human motions conditioned on the given scene and language description. The overall architecture is illustrated in Fig. 3. Specifically, we employ a conditional variational auto-encoder (cVAE) framework to model the probability $p(\Theta_{1:T} \mid S, L_{1:D})$, where $T$ is the length of the motion sequence. Below, we describe the details of each module.

**Condition module** The condition module takes input from two modalities (*i.e.*, the given scene $S$ and the language description $L$) and outputs a joint conditional embedding. To process the 3D scene, we employ the Point Transformer [Zhao et al., 2021] to produce point-level features $[u_1, \cdots, u_{N'}]$, where the transition-down module reduces the cardinality of the scene point cloud from $N$ points to $N'$ points. For the language description $L_{1:D}$, we use the pre-trained BERT [Kenton and Toutanova, 2019] to extract the word-level embedding $[v_1, \cdots, v_T]$.

The point-level features and the word-level features are concatenated and fed into the feature fusion module, a single-layer self-attention module. As a result, both the point-level and the word-level features attend to the features from two modalities by self-attention. The updated point features are concatenated and forwarded to a fully-connected (FC) layer to obtain the scene feature $f_s$. We take the updated [*CLS*] feature as the language feature $f_w$. The scene and language features are finally concatenated and mapped to a conditional latent embedding $z_c$ with FC layers.

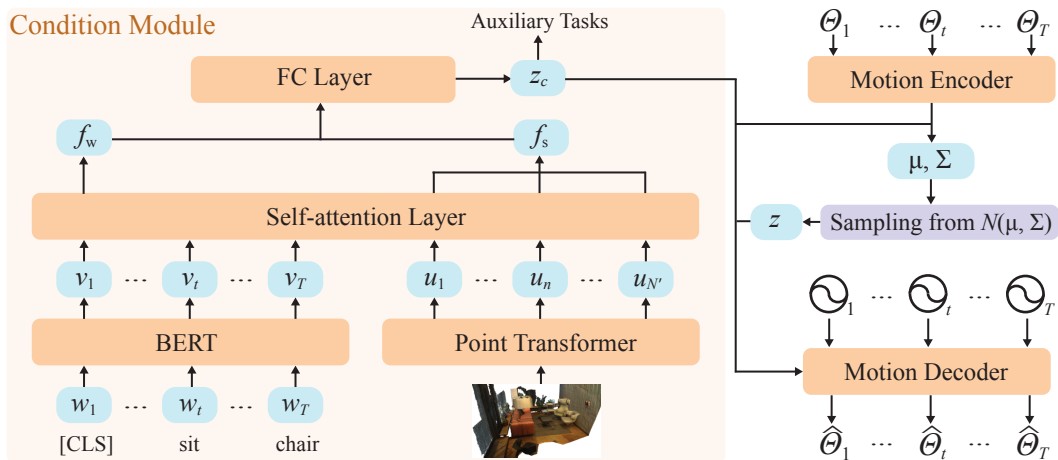

Figure 3: **Illustration of the proposed generative model.** It adopts the cVAE framework and incorporates a motion encoder and decoder to generate human motions. The condition is learned from the joint embedding of the 3D scene and language description with auxiliary tasks of 3D location grounding and action recognition.

**Motion encoder** We first use a bidirectional GRU to obtain a sequence-level feature of the input motion $\Theta_{1:T}$. Next, the output is concatenated with the conditional embedding $z_c$, followed by an MLP layer to predict the Gaussian distribution parameters (*i.e.*, $\mu$ and $\Sigma$). Finally, we sample a latent vector $z$ from the distribution using the reparameterization trick [Kingma and Welling, 2013].

**Motion decoder** Following Petrovich et al. [2021], we use a transformer decoder to generate a sequence of parameters $\widehat{\Theta}_{1:T}$ for a given duration $T$. Specifically, we use $T$ sinusoidal positional embeddings to query the concatenation of the sampled latent $z$ and the conditional embedding $z_c$. The outputs of the transformer decoder are mapped into body meshes with the differentiable SMPL-X model. Of note, although the duration $T$ equals the length of the input motion sequence in the training phase, it can be arbitrary values during inference.

## 4.3 Training loss

We devise the training loss with three components: reconstruction loss, KL divergence loss, and two auxiliary losses for object grounding and action-specific generation. Formally, the training loss is:

$$\mathcal{L} = \mathcal{L}_{rec} + \alpha_{kl}\mathcal{L}_{kl} + \alpha_o\mathcal{L}_o + \alpha_a\mathcal{L}_a, \tag{1}$$

where $\alpha$ denotes the loss weight of each term. We detail each component below.

**Reconstruction loss** Our reconstruction loss is formulated as

$$\mathcal{L}_{rec} = \mathcal{L}_t + \alpha_r\mathcal{L}_r + \alpha_p\mathcal{L}_p + \alpha_v\mathcal{L}_v, \tag{2}$$

where $\mathcal{L}_t$, $\mathcal{L}_r$, and $\mathcal{L}_p$ are $\ell_1$ distance between the predicted body parameters and the ground truth (*i.e.*, global translation $t$, global rotation $r$, and joint rotations $p$). $\mathcal{L}_v$ is an additional term for regressing the positions of the vertices on the SMPL-X body mesh. In practice, the vertex set can be all 10475 vertices on the SMPL-X body mesh or selected sparse markers [Zhang et al., 2021].

**KL divergence loss** Denoting the learned distribution of latent $z$ as $\psi = \mathcal{N}(\mu, \Sigma)$, we regularize $\psi$ to be similar to the normal distribution $\phi = \mathcal{N}(0, \mathbf{I})$, *i.e.*, $\mathcal{L}_{kl} = D_{KL}(\psi \| \phi)$.

**Auxiliary loss** Learning the conditional latent embedding $z_c$ is crucial to our generative model, as it represents compound information of the action, the interacting object, and the 3D scene context from two different modalities. Rather than simply maximizing the negative evidenced lower bound (ELBO), we introduce two auxiliary tasks for locating the interacting target object and generating action-specific motions.

For the first auxiliary task, we directly use an FC layer on top of the latent $z_c$ to regress the target object's center. For the second auxiliary task, we use another FC layer followed by softmax to map the latent $z_c$ to the probability of each action. Specifically, we use the MSE loss $\mathcal{L}_o$ in the target object center regression and the cross-entropy loss term $\mathcal{L}_a$ in the action category classification.

### 4.4 Implementation details

We train our generative model on *HUMANISE* for 150 epochs using Adam [Kingma and Ba, 2014] and a fixed learning rate of 0.0001. For hyper-parameters, we empirically set $\alpha_{kl} = \alpha_o = 0.1$, $\alpha_a = 0.5$, $\alpha_r = 1.0$, and $\alpha_p = \alpha_v = 10.0$. We set the dimension of global condition latent $z_c$ to 512 and latent $z$ to 256. The hidden state size is set to 256 in the single-layer bidirectional GRU motion encoder. The transformer motion decoder contains two standard layers with the 512 hidden state size. We train our model with a batch size of 32 on a V100 GPU. It takes about 50 hours to converge when training the action-agnostic model on the entire *HUMANISE* dataset.

**Point transformer**  Our point transformer module that processes the scene point cloud adopts from the original architecture [Zhao et al., 2021]. For each scene, we down-sample $N = 32768$ points from the original scanned scene and obtain $N' = 128$ points with a 512-D feature for each point after applying the point transformer. We pre-train the point transformer with a semantic segmentation task on ScanNet [Dai et al., 2017] dataset, whose train-test split is the same as *HUMANISE*.

**BERT**  We employ the official pre-trained BERT [Kenton and Toutanova, 2019] model to process the language descriptions with a maximum length of 32. We map the obtained 768-D word-level features into 512-D before concatenating them with the point features.

## 5 Experiments

We benchmark our proposed task—*language-conditioned human motion generation in 3D scenes*—on *HUMANISE* and describe the detailed settings, baselines, analyses, and ablative studies.

### 5.1 Experimental setting

To evaluate the model performance, we conduct experiments in both the action-specific and action-agnostic settings. In the action-specific setting, we train and evaluate our model on *HUMANISE* subsets of each action without the auxiliary task of classifying the action category. In the action-agnostic setting, we train our full model on the entire dataset with auxiliary tasks.

**Dataset splitting**  We split motions in *HUMANISE* according to the original scene IDs and split in ScanNet [Dai et al., 2017], resulting in 16.5k motions in 543 scenes for training and 3.1k motions in 100 scenes for testing.

**Ablative baselines**  We compare our model with three types of ablative baselines to verify the significance of the auxiliary tasks, the feature fusing module, and the scene encoder, respectively:

- We remove the auxiliary tasks for locating the target interacting object and recognizing the action category, denoted as *w/o $\mathcal{L}_o$* (only removing $\mathcal{L}_o$), *w/o $\mathcal{L}_a$* (only removing $\mathcal{L}_a$), and *w/o aux. loss* (removing both $\mathcal{L}_o$ and $\mathcal{L}_a$). We train these models on the entire *HUMANISE* dataset.
- Instead of feeding the scene and the language feature into the self-attention layer, we directly concatenate them as the global conditional feature. We denote this model as *w/o self-att*. We train and test on the *walk* subset.
- We replace the point transformer with PointNet++ [Qi et al., 2017] (*i.e.*, *PointNet++ Enc.*), also pertained with the semantic segmentation task. We train and test on the *walk* subset.

### 5.2 Evaluation metrics

**Reconstruction metrics**  Following Wang et al. [2021a], we introduce the following metrics to evaluate the model's reconstruction capability. We report the $\ell_1$ error ($\times 100$) between the predicted SMPL-X parameters and the ground truth, including global translation $t$, global orientation $r$, and body pose $p$. We also compute the Mean Per Joint Position Error (MPJPE) [Ionescu et al., 2013] and Mean Per Vertex Position Error (MPVPE) in millimeters between the reconstructed SMPL-X body mesh and the ground truth to evaluate the reconstruction in a more intuitive and fine-grained manner. We report these metrics by averaging over the sequence due to different motion duration.

**Generation metrics**  As we hope our generated motion interacts with the correct object specified by the language description, we introduce a body-to-goal distance metric (*goal dist.*). It is the shortest distance (in meters) from the target object to the interacting human body, computed as $\text{Max}(\text{Min}(\Psi^+_{\mathcal{M}}(\mathcal{O})), 0)$, where $\Psi^+_{\mathcal{M}}(\cdot)$ is the positive signed distance field of body mesh $\mathcal{M}$, and

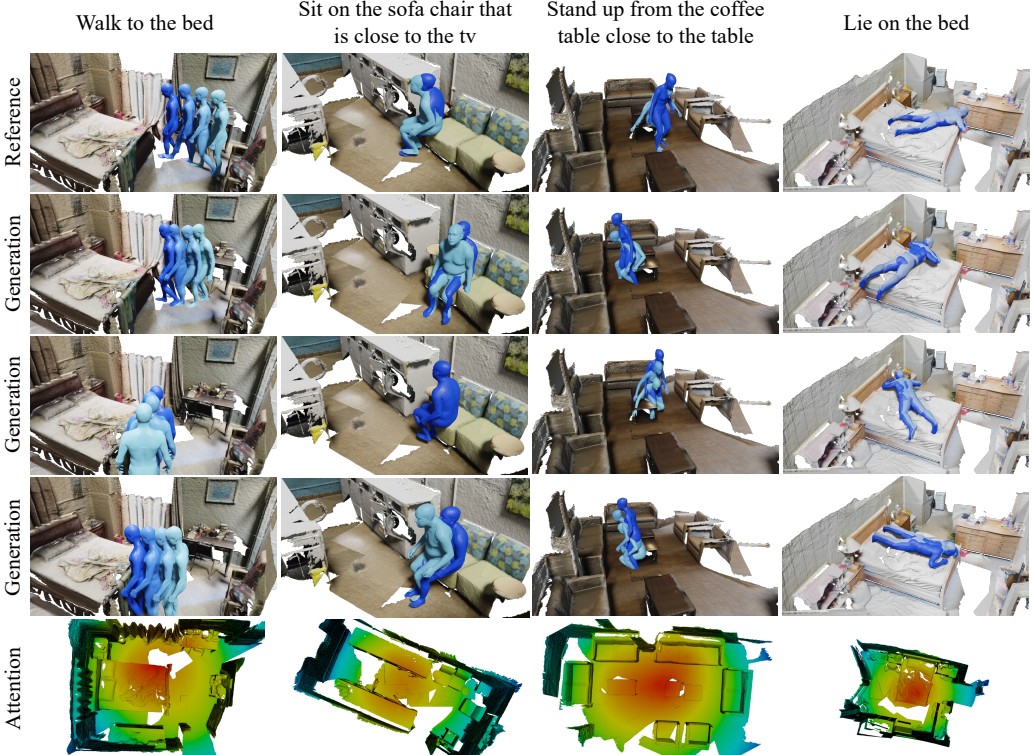

Figure 4: **Qualitative generation results of action-specific models on the *HUMANISE* dataset.** We visualize one reference motion and three generation motions for each scenario. The attention map visualizes the attention weights the $[CLS]$ token attended to the scene point cloud in the self-attention layer. Red denotes higher weight.

$\mathcal{O} \subset S$ the target object point set. Of note, we use the human body in the first frame to compute the body-to-goal distance for *stand up* action and use the last frame for other actions. We generate ten randomly sampled motion sequences for each case in the test split and report the average distance. To measure the diversity of the generated motion, we report the Average Pairwise Distance (APD) [Yuan and Kitani, 2020; Zhang et al., 2021]: The average $\ell_2$ distance between all pairs of motion samples within scenarios. APD is computed as $\frac{1}{K(K-1)T} \sum_{i=1}^{K} \sum_{j \neq i}^{K} \sum_{t=1}^{T} \|\mathbf{x_{i,t}} - \mathbf{x_{j,t}}\|$, where $K$ is sample size (we set $K = 20$), and $\mathbf{x_{i,t}}$ the sampled marker-based representation [Zhang et al., 2021].

**Perceptual study** We perform human perceptual studies to evaluate the generation results in terms of the overall quality and action-semantic accuracy. Given the rendered motion and the language description, a worker is asked to score from 1 to 5 for (i) the overall generation *quality*, and (ii) whether the generated motions are performing the *action* specified by the description. A higher value indicates that the generated result is more plausible with the given scene and language description. We randomly generate samples in 20 scenarios for each model, and three workers score each sample.

## 5.3 Results and discussion

Tab. 2 tabulates the quantitative results of both reconstruction and generation. Fig. 4 and 5 show some qualitative results. Below, we discuss some significant observations.

- As seen in Tab. 2, our model performs well in the action-specific setting, *e.g.*, *walk*, *sit*, and *stand up*. Fig. 4 also shows our model generates human motions that are visually appealing and semantically consistent with the language description. Furthermore, the attention visualizations show that the $[CLS]$ token attends to the point cloud features around the target interacting objects. Meanwhile, we also notice that the *goal dist.* in *lie down* is much smaller than other actions. We suppose the model finds it easier to ground the target object in the *lie down* action, as the interacting objects are mostly large furniture with a flat surface such as "bed" and "table." We provide more clarifications with ablative experiments on the grounding loss weight and data efficiency in Appendix F.

Table 2: **Quantitative results of reconstruction and generation on *HUMANISE* dataset.**

| Model | Reconstruction | | | | | Generation | | | |
|---|---|---|---|---|---|---|---|---|---|
| | transl.↓ | orient.↓ | pose↓ | MPJPE↓ | MPVPE↓ | goal dist.↓ | APD↑ | quality score↑ | action score↑ |
| sit | 5.17 | 3.19 | 1.77 | 113.28 | 112.43 | 0.903 | 10.12 | 2.37±0.85 | 3.79±1.17 |
| stand up | 5.63 | 3.43 | 1.69 | 126.05 | 124.84 | 0.802 | 9.57 | 2.83±1.23 | 4.20±0.94 |
| lie down | 6.46 | 3.09 | 0.76 | 136.87 | 136.20 | 0.196 | 9.18 | 2.31±1.08 | 2.85±1.31 |
| walk | 5.84 | 2.80 | 1.85 | 125.05 | 123.88 | 1.370 | 12.83 | 2.91±1.27 | 3.88±1.26 |
| w/o self-att. | 5.72 | 2.65 | 1.85 | 122.19 | 120.81 | 1.500 | 13.28 | 2.88±1.14 | 3.80±1.09 |
| PointNet++ Enc. | 5.81 | 2.64 | 1.81 | 124.67 | 123.69 | 1.444 | 12.61 | 2.80±1.35 | 3.75±1.27 |
| all actions | 4.20 | 2.91 | 1.96 | 98.01 | 96.53 | 1.008 | 11.83 | 2.57±1.20 | 3.59±1.38 |
| w/o $\mathcal{L}_o$ | 4.20 | 2.89 | 1.93 | 98.15 | 96.69 | 1.383 | 15.09 | 2.42±1.21 | 3.57±1.38 |
| w/o $\mathcal{L}_a$ | 4.23 | 2.91 | 1.95 | 98.67 | 97.11 | 1.135 | 12.66 | 2.17±1.04 | 2.29±1.43 |
| w/o aux. loss | 4.28 | 2.99 | 1.92 | 99.30 | 97.80 | 1.361 | 15.18 | 1.97±0.98 | 2.44±1.38 |

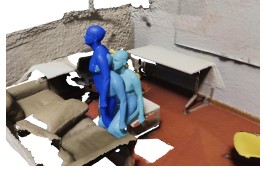 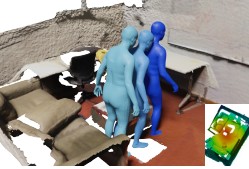 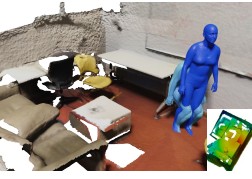 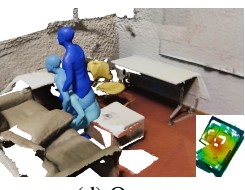

(a) Reference  (b) *w/o aux. loss*  (c) *w/o aux. loss*  (d) Ours

Figure 5: **Ablation results of action-agnostic models.** For the description *sit on the coffee table*, the model *w/o aux. loss* struggles in (b) generating the action specified by the description or (c) locating the interacting object. (d) In comparison, our full model generates motions semantically consistent with the language description.

- The action-agnostic setting is more complex and difficult than the action-specific setting, but the auxiliary tasks improve the model's capability in spatial and action grounding. As shown in Tab. 2, removing $\mathcal{L}_o$ leads to a higher *goal dist.*, which means $\mathcal{L}_o$ can help our generative model to locate the target 3D object. $\mathcal{L}_a$ will help the model generate motions with the correct action label as can be seen from the *action score*. The significant differences in *goal dist.*, *quality score*, and *action score* indicate that the two proposed auxiliary tasks help provide better generation results while maintaining the reconstruction quality. Qualitative results in Fig. 5 further verify this observation.

- The ablative experiments on the feature fusing module and scene encoder validate our model designs. For the action *walk*, our full model outperforms the model *w/o self-att.* in all generation metrics. We conclude that the transformer layers learn better conditional embedding by attending to features across the scene and language modalities. The difference on *goal dist.* compared with *PointNet++ Enc.* shows our better grounding performance through finer-grained scene understanding.

**Discussion of failure cases**  We provide some typical examples of failure cases in Appendix E.

Our model sometimes fails to ground the correct object for interaction as locating objects in the 3D scene with referential descriptions is still challenging [Roh et al., 2022; Chen et al., 2020]. This calls for more efforts in joint learning of different tasks and modalities in language modeling, scene understanding, and motion generation.

Another typical failure case is the incorrect HOI relation with physical implausibility and collision. We suspect this is primarily due to the lack of explicit HOI modeling. Our current architecture is an end-to-end generative model; the scene information is represented by the point-level and scene-level features rather than object-level geometry and semantics. These point out that goal-oriented motion generation may benefit from future work on 4D HOI modeling and object affordance understanding.

**Qualitative results of different duration** $T$    Following Petrovich et al. [2021], we use a transformer decoder as the motion decoder to generate a sequence of body parameters $\widehat{\Theta}_{1:T}$ in a given duration $T$. Fig. 6 shows some generated motions of different duration (*i.e.*, 30 frames, 60 frames, 90 frames, and 120 frames). We render a human body every 15 frames for fair comparisons.

Please refer to Appendix F for additional experiments and Appendix D for more qualitative results.

### 5.4  *HUMANISE* for downstream tasks

We further demonstrate the quality of *HUMANISE* by another existing downstream task proposed in Wang et al. [2021a]: *human motion synthesis in 3D scenes*. This task only considers the scene-conditioned generation, and the language annotations in *HUMANISE* are not used in this setting. More specifically, this task takes a scene point cloud, the start and the end locations, and the start and

| 30 Frames | 60 Frames | 90 Frames | 120 Frames |

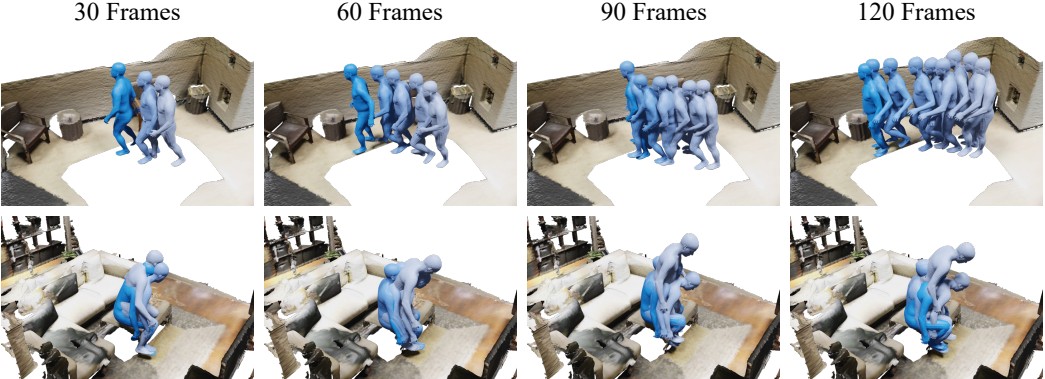

Figure 6: **Motions generated by sampling with different duration** $T$. The language descriptions in these two cases are *walk to the end table that is farthest from the door* and *sit on the coffee table*. Our model is capable of generating motions with various duration.

the end human body poses as input; the goal is to predict all human poses in between. For model architecture, we follow Wang et al. [2021a] to include a RouteNet and a PoseNet. The RouteNet takes the scene point cloud, the start and the end locations and orientations as inputs; it generates a possible route between the start and end location. The PoseNet takes the same scene point cloud, the start and the end human body poses, and the route predicted by RouteNet as inputs; it outputs the parameters for all the human poses along the route.

We train the RouteNet and PoseNet using *HUMANISE* for ten epochs and fine-tune them using PROX [Hassan et al., 2019] for another ten epochs. For a fair comparison, we reproduce the results of the same model trained on PROX for 20 epochs under the same setting. We evaluate both models on the test split of PROX. The reconstruction errors with metrics from Wang et al. [2021a] are reported in Tab. 3. The model pre-trained on *HUMANISE* outperforms the model trained only with PROX on all evaluation metrics by a large margin. This result shows our dataset can alleviate the limitations of existing HSI datasets and potentially support more applications in HSI related topics.

Table 3: **Results of human motion synthesis on PROX dataset.**

| Method | translation | orientation | pose | MPJPE | MPVPE |
|---|---|---|---|---|---|
| [Wang et al., 2021a] | 7.65 | 9.38 | 44.52 | 242.50 | 222.13 |
| *HUMANISE* pre-trained | **6.54** | **8.80** | **42.54** | **222.65** | **201.92** |

# 6 Conclusion

In this work, we propose a large-scale and semantic-rich HSI dataset, *HUMANISE*. It contains diverse and physically plausible interactions equipped with language descriptions. *HUMANISE* enables a new task: *language-conditioned human motion generation in 3D scenes*. We further design a scene-and-language conditioned generative model that produces diverse and semantically consistent human motions based on *HUMANISE*.

**Limitations**    Our work is primarily limited to two aspects. (i) The human motions are relatively short since action-specific motions usually last within 5 seconds. A large-scale, long-duration HSI dataset is still in need for the scene understanding community. (ii) The problem of *language grounding in 3D scenes* is critical for our task but far from being solved, as also revealed in recent 3D grounding works [Achlioptas et al., 2020; Thomason et al., 2022]. A robust grounding module could help generate more meaningful human motions based on language description.

**Acknowledgments and disclosure of funding**    We sincerely thank Chao Xu and Nan Jiang for their valuable advice on this project. We would like to thank the anonymous reviewers for their constructive comments. Z. Wang, Y. Chen, T. Liu, and S. Huang were supported in part by the National Key R&D Program of China (2021ZD0150200); Z. Wang and W. Liang were supported in part by the National Natural Science Foundation of China (NSFC) (No.62172043).

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
