# OpenReview forum: "HUMANISE: Language-conditioned Human Motion Generation in 3D Scenes"
_NeurIPS.cc/2022/Conference — NeurIPS 2022 Accept_

### Official Review · Reviewer_yyeu · 2022-07-10

**Rating:** 7
**Confidence:** 5
**Soundness:** 3 good
**Presentation:** 4 excellent
**Contribution:** 3 good

**Summary:**

The key motivation behind this work is animating human body motion in affordance with the environment ( in this case, 3d indoor scenes), where these animations are grounded semantically in natural language. A precursor to this challenging task is the curation of a large-scale dataset with aligned 3d scenes, human motion and language. The authors have made contributions in the curation of such a dataset named HUMANISE, and an approach to generated diverse and semantically consistent human motions given the 3d scene and language.

**Questions:**

- L123-124: How are the interesting target objects sampled based on the action label?
- Rest of the questions are in the previous section

**Limitations:**

The limitations have been discussed at length in the paper, barring one on trade-off between low-level nuances and data curation simplicity (more in the weaknesses section).

**Strengths And Weaknesses:**

### Strengths
- Generating animation in affordance with environments is a useful and challenging problem. How a person behaves in the environment can depend completely on (1) high level conditions such as how the objects are arranged in the environment and also on (2) low-level conditions such as the subtle differences between the size of the chair, or the hand placement depending on where a person sits on a big couch. Addition of language to this challenging problem is a welcome addition and simplicity of the data curation is commendable. I hope that the data collection scripts would be released for the community to use.
- Animation generation in context of scenes and language is complex. To that end, both quantitative and qualitative experiments were performed which make the analysis more solid.

### Weaknesses
- The descriptions are generated using a fixed template and four actions. That can be quite limiting to the diversity of the possible sentences, especially in out-of-domain language contexts. How does the model behave in those scenarios
* Consider the example about "sit on the armchair near the desk" example used in the Introduction. While the semantics of finding the right chair and sitting down on it are the high level conditions been looked after, low level nuances (discussed in [1]) are ignored here such as the positions of the hands. Are they on the armrests or are they on the table or something else?
* This brings me to a concern about the construction of the dataset. As existing animations are aligned with 3d scenes, low-level nuances will likely be incorrect. While I understand that this is a trade-off of simple data curation vs low-level accuracy, how much do we lose when we don't care about these low-level nuances?
* How is the training and test split done? For example, consider "sit on the coffee table" in the train set. Is there a semantically similar sentence in the test set such as "sit on the coffee table near the couch"? If so, I would be curious to see some analysis of cases where this is not true because they can be a plausible edge-case (or out-of-domain scenario).
* I would like to put forth one argument (not necessarily a weakness) on which I hope to have a constructive discussion with the authors. Let's assume we only care about the high-level semantics. This reduces the problem of first finding the target object (from the 3d scene and language), followed by figuring out the correct action to perform (from the language) and finally use these two pieces of information to generate a plausible animation (as discussed in the model section of the paper). Finding the target object using natural language and 3d scenes is a relatively well studied problem. Determining the action from the language is relatively straightforward given that the language sentences are template based and there are only 4 actions. And finally target based animation is also a well-studied problem in the graphics community. Where does the technical novelty of this paper lie? Are the authors positioning the paper that combines these three ideas to solve a more complex problem, or is there something I am missing out?

#### Minor Suggestions
- In figure 4 column 2, where is the TV. Some of the images are not very clear to make out the details.
- Table 2: the \hline is probably intended between walk and w/o self-attn.

[1] - Starke, Sebastian, et al. "Neural state machine for character-scene interactions." _ACM Trans. Graph._ 38.6 (2019): 209-1.

---

> ### Author Response · Authors · 2022-08-02
> **Response to Reviewer yyeu**
>
> Thank you very much for your valuable feedback! We sincerely hope that our response can address your concerns.
>
> ### 1. Concerns about language diversity and generalization to out-of-domain language.
>
> Please refer to the general responses.
>
> ### 2. The trade-off between data curation efficiency and low-level nuances.
>
> This is a very good observation about the ``low-level nuances'' in HSI research. The detailed hand-object interaction is not considered in our current data generation pipeline. For one thing, current HSI research emphasizes the high-level semantic and physical plausibility, also our focus in this paper. For another, how to deal with the detailed part-level interaction between humans and objects is still an open and ongoing research topic. Both Neural-State Machine [2] and SAMP [3] assume limited action types and familiar object geometry to predict the contact point. Even though low-level constraints are not extensively optimized in our pipeline, the quantitative human study (Tab. 1 in Supp. Mat.) demonstrates HUMANISE still has higher quality in terms of collision, smoothness, HSI, and overall quality compared to PROX. This shows that our dataset is a good extension of the existing HSI datasets and achieves a reasonable and practical balance between data curation efficiency and low-level nuances.
>
> ### 3. Clarification of the train-test split. Are there semantically different sentences in the training and test set?
>
> We split HUMANISE according to the original scene IDs in ScanNet, i.e., HSIs with scene IDs less than 600 are in the training set, and HSIs with scene IDs greater than 600 are in the test set. There is no clear split standard in terms of the language description; most sentences in the test set will have semantically similar ones in the training set. However, this task is still challenging as it requires more than language understanding, i.e., the accurate grounding in 3D unseen scenes and scene-aware plausible motion generation. This is similar to Sr3D [1], where the 3D grounding task is challenging despite the template-based simple descriptions.
>
> ### 4. Discussion about decomposing the task into sub-tasks and the technical novelty of this paper.
>
> Conceptually, solving the language-conditioned human motion generation task in 3D scenes can be decomposed into three sub-tasks: language grounding, action determination, and motion generation. However, it does not necessarily mean that solving the three sub-tasks independently can solve the more complex task since these sub-tasks are not mutually independent and the error will accumulate across sub-tasks. A similar case is the one-stage/two-stage visual language grounding. One-stage methods generally outperform two-stage methods, in which object detection and language grounding are performed separately.
>
> Our technical novelty mainly lies in (1) the first step to handle the proposed new task with an end-to-end baseline and (2) the incorporated auxiliary loss functions inspired by the aforementioned ideas of how to address the joint task. Experiments show that the proposed method handles the task well and the auxiliary losses indeed improve the performance.
>
> ### 5. How to sample the target object based on the action label?
>
> We predefined the potential object categories for each action type. Given an action label, the interacting object is sampled randomly from all the objects in the scene that fall into those categories.  We will clarify this in our revision.
>
> ### 6. The quantitative results grouping in Tab. 2.
>
> We group the quantitative results in Tab. 2 so that the result comparison will contribute to the discussion in Sec. 5.3 in the main paper. We will clarify this in the revision.
>
> ### Bibliography
>
> [1] Panos Achlioptas, Ahmed Abdelreheem, Fei Xia, Mohamed Elhoseiny, and Leonidas Guibas. Referit3d: Neural listeners for fine-grained 3d object identification in real-world scenes. In _European Conference on Computer Vision (ECCV)_, 2020.
> [2] Sebastian Starke, He Zhang, Taku Komura, and Jun Saito. Neural state machine for character-scene interactions. _ACM Transactions on Graphics (TOG)_, 38(6):209–1, 2019.
> [3] Mohamed Hassan, Duygu Ceylan, Ruben Villegas, Jun Saito, Jimei Yang, Yi Zhou, and Michael J Black. Stochastic scene-aware motion prediction. In _International Conference on Computer Vision (ICCV)_, 2021.

---

> > ### Comment · Reviewer_yyeu · 2022-08-08
> > **Response to reviewer**
> >
> > Thank you for the responses. I have read all the reviews and responses. I am satisfied with the responses of my questions.
> >
> > Overall, I think this is a very ambitious direction, and the authors have done a good job of providing the community with a valuable dataset and starting point to explore further. Hence, I am raising my score by 1

---

> ### Author Response · Authors · 2022-08-08
> **Response to Reviewer yyeu**
>
> Dear reviewer:
>
> Thanks again for your constructive suggestions, which have helped us improve the quality and clarity of the paper!
>
> Since the discussion phase is about to end, we have not heard any post-rebuttal response yet.
>
> Please don’t hesitate to let us know if there are any additional clarifications or experiments that we can offer, as we would love to convince you of the merits of the paper. We appreciate your suggestions. Thanks!

---

### Official Review · Reviewer_1Evw · 2022-07-11

**Rating:** 6
**Confidence:** 4
**Soundness:** 2 fair
**Presentation:** 4 excellent
**Contribution:** 2 fair

**Summary:**

This work proposes a new dataset containing paired human motion, scene semantic labels, and motion language annotation. Its main contribution lies in an automatic pipeline for generating human-scene interactions that are both semantically and language labeled. Leveraging a large-scale motion dataset, semantically labeled 3D scenes, rule-based instruction generation, and collision/contact constraints, this dataset contains better quality motion and human-scene interactions than previous video-based datasets (e.g. PROX). Leveraging this dataset, a generative model is proposed to tackle the new task, language and scene-conditioned 3D human motion generation.

**Questions:**

**Dataset Generation and motion diversity**

- What is the motion diversity and number of samples for each action? BABEL provides action labels for the AMASS dataset, but for actions such as walking and sitting up and down, there are more samples than lying on beds. For the 4k motion sequences, are they all motion sequences of different action/motion? What is the data distribution among the actions?

**Qualitative Results**

- In the provided video, at timestamp 6:48, the right bottom motion does not correspond to any meaningful human motion.
- Quantitative comparison with PROX is not entirely fair and does not really provide additional insight. The motion sequences from PROX are recovered from optimization-based methods and is not for the purpose of human motion generation (more for pose estimation). It is bound to have lesser quality than MoCap sequences from AMASS dataset.

**Limitations:**

Yes, the authors have addressed the limitations adequately.

**Strengths And Weaknesses:**

## Strength

**Automatic data generation**

- The main strength of this work lies in the proposed dataset and automatic data generation pipeline. The idea of using existing MoCap data from AMASS and scene scans to create paired human scene interaction is intuitive and interesting. Such a pipeline does not rely on any additional data capture equipment and can help reuse existing datasets for HSI-related tasks. The proposed contract and collision constraints can largely alleviate the violation of physical constraints in this automatic data generation framework.

**New Task: scene and language conditioned motion generation**

- While language/action-conditioned human motion generation [1,2], scene-conditioned motion generation [3], and goal-conditioned motion generation [4] has been separately studied to a certain extent, language ***and*** scene-conditioned human motion generation has not been explored as much due to the lack of annotated dataset. All the challenges that separately exist in the above tasks will be more apparent in these combined tasks (motion realism, physical validity, multi-modal nature of human motion). This paper proposed a general baseline for further studying this interesting and impactful task.
- The proposed VAE network, though similar to prior arts in conditioned human motion generation [1,2] does serve as a decent baseline for future works in this task. The auxiliary task loss is a viable approach to gain better performance.

## Weakness:

**Dataset Creation Methodology:**

- While the created dataset is certainly useful for future HSI research, the limited action types and motion diversity (see in questions) takes away some of the novelty. The proposed framework can be viewed as fitting a scene for existing motion sequences, and largely relies on the action labels for matching the motion sequences. It is more interesting to see how diverse the **generated** motion from learning-based methods, in the hope that learning-based methods and learn from a few interaction samples and actually learn affordance.

**Auxiliary task loss:**

- While the auxiliary task is effective in boosting motion reconstruction performance, it takes away the language conditioning and more or less makes the task degenerates into an action classification and goal-reaching task.

**Qualitative Results:**

- More qualitative evaluation is needed to fully evaluate the quality of the dataset and the generative methods. Since motion is best seen in videos, it is important to include more quantitative results for both the dataset and the generative method for better evaluation. It is easy to pick sequences that fit the current text description, but the overall dataset quality is hard to judge based on a handful of samples.

**Motion Diversity:**

- For a motion generation task, it is important to include evaluation metrics on motion diversity. While the included metrics such as goal distance and action score are important for scene and language conditioning, given the existence of the auxiliary task loss, the proposed method can easily memorize a few motions that match goal and actions.

## Small issues:

- L2: mediocore “characteristics”
- L87: study HSI related topics

[1] Petrovich, Mathis, Michael J. Black and Gül Varol. “Action-Conditioned 3D Human Motion Synthesis with Transformer VAE.” *2021 IEEE/CVF International Conference on Computer Vision (ICCV)* (2021): 10965-10975.

[2] Ahuja, Chaitanya and Louis-Philippe Morency. “Language2Pose: Natural Language Grounded Pose Forecasting.” *2019 International Conference on 3D Vision (3DV)* (2019): 719-728.

[3] Cao, Zhe, Hang Gao, Karttikeya Mangalam, Qi-Zhi Cai, Minh Vo and Jitendra Malik. “Long-term Human Motion Prediction with Scene Context.” *ArXiv* abs/2007.03672 (2020): n. pag.

[4]Hassan, Mohamed et al. “Stochastic Scene-Aware Motion Prediction.” *2021 IEEE/CVF International Conference on Computer Vision (ICCV)* (2021): 11354-11364.

---

> ### Author Response · Authors · 2022-08-02
> **Response to Reviewer 1Evw**
>
> Thank you very much for your valuable feedback! We sincerely hope that our response can address your concerns.
>
> ### 1. Concerns over dataset generation and motion diversity.
>
> Please refer to the general responses for clarification.
>
> **Detailed motion statistics:** For the motion diversity, 1268 different motions from AMASS are selected to synthesize 4892 motion sequences in 512 3D scenes. Among all the synthesized motion sequences 4892(1268), the number of the synthesized motion sequences and the number of selected motions from AMASS for each action are: 2488(715) for `walk`, 1299(355) for `sit`, 936(165) for `stand up`, and 169(33) for `lie`. Please refer to Sec. 3.3 in the main paper and Sec. A in the Supp. Mat. for more dataset statistics.
>
> To measure the diversity of the generated motion, we report the Average Pairwise Distance (APD) [3,4,5], i.e., the average L2 distance between all pairs of motion samples within scenarios. APD is computed as $\frac{1}{K(K-1)}\sum_{i=1}^{K}\sum_{j\neq i}^{K}||\mathbf{x_i} - \mathbf{x_j}||$, where $K$ is sample size and $\mathbf{x_i}$ is the sampled marker-based representation[5] of pose sequence. Here we choose $K=20$; the unit is meter.
>
> |Model|sit|stand up|lie down|walk|w/o self-att.|PointNet Enc.|all actions|w/o aux. loss|
> |:-|:-|:-|:-|:-|:-|:-|:-|:-|
> |APD|9.92|7.49|7.71|6.76|10.36|13.16|10.54|10.17|
>
> From the table, we can see the generated motions are quite diverse. It also can be seen in Fig. 4 in the main paper and Fig. 5 in the Supp. Mat. that our model is able to generate multiple meaningful motions for each scenario.
>
> ### 2. Whether the auxiliary tasks degenerate the proposed task into an action classification and goal-reaching task.
>
> The auxiliary tasks we propose in the framework act as the inductive bias to help the model better understand the language and perform 3D grounding, as can be seen from the ablation study. Since our method is an end-to-end model without stages, the task does not degenerate into sub-tasks. In contrast, prior work like [1,3] actually decouples the task into sub-tasks of action generation and goal-reaching.
>
> ### 3. More qualitative results to evaluate the quality of the dataset and the generative methods.
>
> The Supp. Mat. and the demo video provides some data samples and qualitative results. We also provide additional examples in video form from the [anonymous website](https://dsdbhj.github.io/humanise_material/index.html#dataset). We will further design an explorer upon the release of the dataset.
>
> ### 4. Quantitative comparison with PROX is not entirely fair.
>
> Although PROX was initially proposed for pose estimation, it is now the most popular dataset for HSI research and motion generation. The gap between the estimated human poses in PROX and the MoCap sequences is exactly the motivation of our work, i.e., proposing a large-scale, high-quality synthetic dataset to facilitate the research on scene-conditioned human motion generation. The comparison between our dataset and PROX justifies our motivation.
>
> ### 5. Implausible motion in the demo video at timestamp 6:48.
>
> This motion is a failure case generated by the lie-down action model. The worse results in the `lie down` action might be due to the unbalanced data distribution (the lie-down subset is only $10\%$ of the sit subset).
>
> ### Bibliography
>
> [1] Mohamed Hassan, Duygu Ceylan, Ruben Villegas, Jun Saito, Jimei Yang, Yi Zhou, and Michael J Black. Stochastic scene-aware motion prediction. In _International Conference on Computer Vision (ICCV)_, 2021.
> [2] Mohamed Hassan, Vasileios Choutas, Dimitrios Tzionas, and Michael J Black. Resolving 3d human pose ambiguities with 3d scene constraints. In _International Conference on Computer Vision (ICCV)_, 2019.
> [3] Kaifeng Zhao, Shaofei Wang, Yan Zhang, Thabo Beeler and Siyu Tang. Compositional Human-Scene Interaction Synthesis with Semantic Control. In _European Conference on Computer Vision (ECCV)_, 2022.
> [4] Ye Yuan and Kris Kitani. Dlow: Diversifying latent flows for diverse human motion prediction. In _European Conference on Computer Vision (ECCV)_, 2020.
> [5] Yan Zhang, Michael J Black, and Siyu Tang. We are more than our joints: Predicting how 3d bodies move. In _Conference on Computer Vision and Pattern Recognition (CVPR)_, 2021.

---

> > ### Comment · Reviewer_1Evw · 2022-08-07
> > **Follow up to author response**
> >
> > Thanks for the detailed response!
> >
> > My remaining question still centers on the auxiliary task and on a bigger scale, how general the proposed pipeline can be. I feel like the auxiliary task and the necessity of the auxiliary task prove my point that the task is degenerating action classification and goal-reaching. The action class and goal location are now explicitly involved in the loss function and guide the model to become classification and goal-reaching tasks. I feel like a key-world-based action matcher and target object extractor (based on a POS tag or something similar) might be enough to be used as tokens to be fed into the classifier. A BERT embedding does not seem necessary. This way, the task degenerates into an action-conditioned goal-reaching task, rather than "language conditioned motion generation". How well would a baseline that directly uses action-conditioning work?
> >
> > As mentioned in the response to reviewer "yyeu": "However, it does not necessarily mean that solving the three sub-tasks independently can solve the more complex task since these sub-tasks are not mutually independent and the error will accumulate across sub-tasks." I feel like there needs to be more discussion and ablation around this point. Training end-to-end does not always lead to better performance, and claiming "end-to-end" is a better approach in this problem seems overreaching. Training the system end-to-end does not automatically lead to better performance.

---

> > > ### Author Response · Authors · 2022-08-09
> > > **Further response**
> > >
> > > Thanks for further explaining your concerns over the auxiliary tasks and how “language-conditioned motion generation in 3D scene” can be decomposed into sub-tasks. Again, we agree that this complex task can be decomposed into language grounding, action determination, and motion generation, and we are not claiming “end-to-end” is a better approach. In this paper, we mainly intend to provide a feasible baseline for the introduced task, and we hope more efforts can be devoted to exploring this task in the future.
> > >
> > > As an initial attempt to explore the difference between an end-to-end approach and a cascaded approach for our task, we conduct experiments to compare with baselines that directly use action or target object as conditions. More specifically, We modify our CVAE model by replacing the condition with the concatenation of the global scene feature, the target object center, and the action category. We use the point cloud feature extracted from PointNet++ as the global scene feature. We adopt the one-hot embedding to represent the GT action categories. For the target object center, we use a 3D grounding model pre-trained on ScanNet, _i.e._, ScanRefer [1], to estimate the target object center. We denote this baseline as $\mathrm{GT_{action}}$. We also test an ablative baseline that directly uses the ground truth position instead of the predicted target object position. We denote this baseline as $\mathrm{GT_{action+target}}$. The quantitative results are as follows.
> > >
> > > Model       | $\mathrm{transl.}\downarrow$ | $\mathrm{orient.}\downarrow$ | $\mathrm{pose}\downarrow$ | $\mathrm{MPJPE}\downarrow$ | $\mathrm{MPVPE}\downarrow$ | $\mathrm{goal dist.}\downarrow$ | $\mathrm{APD}\uparrow$
> > > -|-|-|-|-|-|-|-
> > > $\mathrm{GT_{action}}$ | 8.76 | 5.83 | 5.44 | 209.06 | 203.33 | 1.305  | 13.39
> > > $\mathrm{GT_{action+target}}$ | 8.06 | 5.85 | 5.32 | 193.89 | 188.78 | 0.246  | 8.83
> > > Ours         | 8.61 | 6.43 | 5.71 | 210.96 | 205.42 | 1.081 | 10.54
> > >
> > > From the table, we can see that the baseline that directly uses GT action-conditioning reaches approximately the same performance as our model, while utilizing the GT action and the GT target position can significantly improve the motion generation metrics. We hypothesize this is because the action can be easily parsed from the instruction and 3D object grounding is significantly more challenging than other subtasks, which affects the performance most. The results also justify the decomposition could achieve similar performance in current complexity, we will clarify this finding in our revised version, thank you!
> > >
> > > [1] Dave Zhenyu Chen, Angel X Chang, and Matthias Nießner. Scanrefer: 3d object localization in rgb-d scans using natural language. In _European Conference on Computer Vision (ECCV)_, 2020.

---

### Official Review · Reviewer_bBY2 · 2022-07-13

**Rating:** 4
**Confidence:** 4
**Soundness:** 3 good
**Presentation:** 2 fair
**Contribution:** 3 good

**Summary:**

This paper released a synthesized dataset for language-conditioned human motion generation in 3D scenes, which aligns the existing 3D human motion datasets in 3D scene datasets to synthesize human motions in 3d scene. Besides, this paper proposes a baseline model based on conditional VAE for the language-conditioned human motion generation in 3D scene. This topic is promising and meaningful.

**Questions:**

1) Can you report some experimental comparisons of your model with other conditional generation models? Why do you use the CVAE model, and what are the effects of other conditional generation models? It is insufficient to judge the advanced property of the CVAE model only by ablation studies.
2) How about the Params and the GFLOPs of your model?
3) Do the dataset and the model proposed in this paper adequately consider the body-scene interactions? In other words, are the motions in the dataset or generated by model appropriate to the environment? For example, human sitting on the couch and sitting on the toilet may have different postures. Simply applying the same motion to different environments may result in unreasonable actions.


**Ethics Review Area:**

["I don’t know"]

**Limitations:**

Authors are advised to provide more and longer-sequence video demonstrations for the proposed dataset.
The body-scene interactions should be considered and refined in the proposed dataset for plausible human motion generation.


**Strengths And Weaknesses:**

Strengths:
A large-scale and semantic-rich synthetic HSI dataset is proposed in this work, which enables a new task: language-conditioned human motion generation in 3D scene.

Weaknesses:
1) Though this is the large-scale and semantic-rich synthetic HSI dataset, the number of the action categories is limitted. There are only four interacitve indoor actions, i.e sit, stand up, lie down, and walk. The rich human-object interactions like opening a refrigerator, close the door are not included;
2) The sequence length of each sample in the proposed dataset seems too short to express rich semantic information;
3) As the motion dataset are aligned, not actually acted, in the 3D scene, some motion in the interaction cannot well presented in the dataset, for example torch a chair before siting on it;

---

> ### Author Response · Authors · 2022-08-02
> **Response to Reviewer bBY2**
>
> Thank you very much for your valuable feedback! We sincerely hope that our response can address your concerns.
>
> ### 1. Action category and motion lengths.
>
> Please refer to the general responses for clarification.
>
> ### 2. Clarification of CVAE-based generation framework.
>
> The CVAE framework is widely used in conditional human motion/pose generation tasks [1,2,3,4,5]. Our experimental and ablation results demonstrate that the proposed framework, the module design, and the auxiliary loss functions are effective for this task. As also pointed out by reviewer 1Evw, we think our proposed framework acts as a decent baseline for future works on this task. We hope more sophisticated conditional generative models can build upon our method in the future.
>
> ### 3. Model size and GFLOPs.
>
> The total model size is about 129.3M, including the BERT module which consumes 109.5M. Due to customized modules, we cannot accurately evaluate the GFLOPs of our model. Alternatively, we report the training time, which partially reflects the computational complexity of the model. It takes about 16 hours to train our model on HUMANISE with a single V100 GPU and a batch size of 24.
>
> ### 4. Are the motions in the dataset or generated by the model appropriate to the environment? Simply applying the same motion to different environments may result in unreasonable actions.
>
> Our data synthesis pipeline pinpoints two critical factors in realistic HSI: semantic consistency and physics plausibility, which are also the focuses and metrics of previous work on human motion generation [1,2,3,5]. We argue that the motions that satisfy these two factors are reasonable and appropriate interactions in 3D scenes. Therefore, human poses for `sit on couch` and `sit on toilet` could be similar. Collecting all possible interactions for certain actions, for example, all human postures of `sit on toilet`, requires significantly larger effort and is not our primary focus.
>
> Furthermore, human study (Tab. 1 in Supp. Mat.) also demonstrates HUMANISE has higher quality in terms of collision, smoothness, HSI, and is more appropriate compared to PROX.
>
> ### Bibliography
>
> [1] Jiashun Wang, Huazhe Xu, Jingwei Xu, Sifei Liu, and Xiaolong Wang. Synthesizing long-term 3d human motion and interaction in 3d scenes. In _Conference on Computer Vision and Pattern Recognition (CVPR)_, 2021.
> [2] Yan Zhang, Mohamed Hassan, Heiko Neumann, Michael J Black, and Siyu Tang. Generating 3d people in scenes without people. In _Conference on Computer Vision and Pattern Recognition (CVPR)_, 2020.
> [3] Mathis Petrovich, Michael J Black, and Gül Varol. Action-conditioned 3d human motion synthesis with transformer vae. In _International Conference on Computer Vision (ICCV)_, 2021.
> [4] Yan Zhang, Michael J Black, and Siyu Tang. We are more than our joints: Predicting how 3d bodies move. In _Conference on Computer Vision and Pattern Recognition (CVPR)_, 2021.
> [5] Mohamed Hassan, Duygu Ceylan, Ruben Villegas, Jun Saito, Jimei Yang, Yi Zhou, and Michael J Black. Stochastic scene-aware motion prediction. In _International Conference on Computer Vision (ICCV)_, 2021.

---

> ### Author Response · Authors · 2022-08-08
> **Response to Reviewer bBY2**
>
> Dear reviewer:
>
> Thanks again for your constructive suggestions, which have helped us improve the quality and clarity of the paper!
>
> Since the discussion phase is about to end, we have not heard any post-rebuttal response yet.
>
> Please don’t hesitate to let us know if there are any additional clarifications or experiments that we can offer, as we would love to convince you of the merits of the paper. We appreciate your suggestions. Thanks!

---

> ### Comment · Reviewer_bBY2 · 2022-08-10
> **thanks for the reply**
>
> Thanks the authors for the responses. I have no further questions.

---

> > ### Author Response · Authors · 2022-08-10
> > **Thanks again for your review**
> >
> > Thanks for your time and constructive comments. We will integrate the feedback into the revision and further improve the quality and clarity of the paper. If we have resolved all your concerns, we kindly ask you to consider raising the rating. We believe our work would promote future research in the community!

---

### Official Review · Reviewer_vzPA · 2022-07-15

**Rating:** 5
**Confidence:** 4
**Soundness:** 3 good
**Presentation:** 3 good
**Contribution:** 2 fair

**Summary:**

The paper focuses on human-scene interaction, and the authors propose a new dataset and a new task of language + scene conditioned synthesis of 3D human motion. Given a 3D scene as RGB point clouds and a language description (“sit on the couch”), the goal is to synthesize realistic and plausible motion in the given scene.

They introduce the HUMANISE dataset, which contains (4892) AMASS 3D motion sequences aligned with (512) ScanNet indoor scenes. The authors do motion alignment of AMASS sequences with the scenes using collision and contact constraints. The language descriptions for these aligned motion sequences are generated using templates.  They propose a reasonable encoder-decoder model to conditionally synthesize human motion sequences and provide clear ablations to back their design.

**Questions:**

Given the dataset, I’m curious to hear the authors' thoughts on how they expect the community to build on it.
1) Difficulty or quality of the dataset: It seems like the dataset is not particularly challenging regarding both language description and length of motions. Given this, the models trained on HUMANISE may never generalize to novel descriptions & scenes.
2) How the dataset generation process is a viable solution to generating rich HSI datasets with language descriptions? This seems to be one possible approach, but I’m not sure it’s a promising one. Finding a motion sequence recorded in isolation to have rich interactions in an unknown scene seems unlikely. To solve the task in the right way, one might actually need to capture motion occurring in scenes (which is a very challenging task in itself).

Minor fixes:
- Table 1 caption: HUAMNISE → HUMANISE
- Line 114: (i.e., ScanNet [Dai et al., 2017] → (i.e., ScanNet [Dai et al., 2017])


**Limitations:**

Authors have discussed limitations and societal impact of their work.


**Strengths And Weaknesses:**

Strengths:
- The problem of 3D motion generation conditioned on language and scene is an interesting and important one. HUMANISE is the first dataset and approach to tackling this.
- The qualitative and quantitive results of the proposed model are good.
- The paper is also clearly written. All the arguments and technical decisions made in the paper are well-backed by reasoning or experiments.

Weakness:

My primary concern is with the dataset itself and the methodology of creating the dataset.

First, most sequences are very short (averaging < 2 sec at 30fps). There are only four action types considered (walk/stand/sit/lie), with “lie” being < 4% of the data. The other action types don’t have rich interactions with the environment (“walk to the table”, “sit on the couch”). Similarly, the language description is not extremely interesting as it mostly composes an action type + object type. The power of language would be in expressing more interesting compositions and interactions.

Given this, I feel the representation in Table 1 is not giving a complete picture. Yes, HUMANISE has many more scenes and clips, but the variation in diversity should also be stated.

Fundamentally these limitations might be due to trying to align a motion to a 3D scene it doesn’t belong. This greatly decreases the possible motions that could be aligned and the possible object and scene interactions that the motion + language can describe (other than walk/sit/stand/jump etc.)

---

> ### Author Response · Authors · 2022-08-02
> **Response to Reviewer vzPA**
>
> Thank you very much for your valuable feedback! We sincerely hope that our response can address your concerns.
>
> ### 1. Concerns over dataset generation and diversity.
>
> Please refer to the general responses about the concern of the action category. Here we provide additional quantitative analysis and comparison of HUMANISE as an extension of Tab. 1 in the main paper. 1268 different motions from AMASS are selected to synthesize 4892 motion sequences in 512 3D scenes. From the per-frame interaction annotation on PROX from [4], $80.0\%$ of the frames belong to the `walk`, `stand`, `sit`, `lie`, where `lie` takes about $6\%$. These results verify that HUMANISE is similar to the existing HSI dataset in terms of motion diversity but further possesses a significant advantage in scene variety.
>
> ### 2. Difficulty or quality of the dataset. Generalization to novel descriptions and scenes for modeled trained on HUMANISE.
>
> Please refer to the general responses to the questions regarding language descriptions and motion lengths. Here we further emphasize the challenge and difficulty of the task, and also the generalization ability of models fostered by our data. As also pointed out by other reviewers (1Evw, yyeu), the challenges of language-conditioned human motion generation in 3D scenes lie in the combination of motion realism, physical plausibility, and multi-modal 3D grounding. We argue our proposed task is much more complicated than previous works [1,2,3], which have already shown to be challenging as individual tasks.
>
> For the generalization capability, our proposed dataset provides the chance to evaluate many more unseen scenes (85 during test time) compared to [2,4] (4 unseen scenes in PROX). Experimental results show that our model trained on HUMANISE can generalize to unseen 3D scenes. Our model can also generalize to motion generation with different durations, as seen from the qualitative results shown in Fig. 6 in Supp. Mat. As mentioned in the general responses, we also test our model's generalization ability on natural descriptions from human annotators in (1) the test split of HUMANISE, and (2) the new dataset Replica [5]. As shown in the [anonymous website](https://dsdbhj.github.io/humanise_material/index.html#generalize), our model can generate meaningful motions under human-annotated descriptions, even in new scenes from other datasets.
>
> ### 3. Viability of generating rich synthetic HSI datasets by the proposed pipeline.
>
> As discussed in the general responses, capturing motions in real scenes is indeed a challenging and labor-consuming task. Thus, producing data with a synthetic pipeline is a viable way to address the limitations of the existing datasets (i.e., quality, scale, scene diversity) and facilitate HSI research. Our pipeline pinpoints two critical factors in realistic HSI: semantic consistency and physics plausibility. Human study (Tab. 1 in Supp. Mat.) demonstrates HUMANISE has higher quality in terms of collision, smoothness, HSI, and overall quality compared to PROX.
>
> ### Bibliography
>
> [1] Zhe Cao, Hang Gao, Karttikeya Mangalam, Qi-Zhi Cai, Minh Vo, and Jitendra Malik. Long-term human motion prediction with scene context. In _European Conference on Computer Vision (ECCV)_, 2020.
> [2] Jiashun Wang, Huazhe Xu, Jingwei Xu, Sifei Liu, and Xiaolong Wang. Synthesizing long-term 3d human motion and interaction in 3d scenes. In _Conference on Computer Vision and Pattern Recognition (CVPR)_, 2021.
> [3] Mohamed Hassan, Vasileios Choutas, Dimitrios Tzionas, and Michael J Black. Resolving 3d human pose ambiguities with 3d scene constraints. In _International Conference on Computer Vision (ICCV)_, 2019.
> [4] Kaifeng Zhao, Shaofei Wang, Yan Zhang, Thabo Beeler and Siyu Tang. Compositional Human-Scene Interaction Synthesis with Semantic Control. In _European Conference on Computer Vision (ECCV)_, 2022.
> [5] Julian Straub, Thomas Whelan, Lingni Ma, Yufan Chen, Erik Wijmans, Simon Green, Jakob J Engel, Raul Mur-Artal, Carl Ren, Shobhit Verma, et al. The replica dataset: A digital replica of indoor spaces. _arXiv preprint arXiv:1906.05797_, 2019.

---

> ### Author Response · Authors · 2022-08-08
> **Response to Reviewer vzPA**
>
> Dear reviewer:
>
> Thanks again for your constructive suggestions, which have helped us improve the quality and clarity of the paper!
>
> Since the discussion phase is about to end, we have not heard any post-rebuttal response yet.
>
> Please don’t hesitate to let us know if there are any additional clarifications or experiments that we can offer, as we would love to convince you of the merits of the paper. We appreciate your suggestions. Thanks!

---

### Author Response · Authors · 2022-08-02
**General Responses**

We thank all the reviewers for their time and valuable comments. Below, we first clarify common concerns and summarize revisions.

## 1. Clarification of action categories

### Design principle

HUMANISE aims to address the limitations of the previous HSI datasets regarding motion quality, scene variety, and semantics. As action diversity is not the topmost focus, we choose the action types by following common practices from previous works [4,6,8,9,10] on human motion generation in 3D scenes, which only deal with `walk`, `sit`, `stand up`, and `lie`. Note that the introduced four actions are also the most common daily indoor activities, dominant in previous HSI datasets, such as PROX [1]. The action type distribution is also long-tailed in AMASS [11] and BABEL [7], in which the open and close actions mentioned by reviewers only have 19 and 6 segments, respectively.

### Generate other action types

We have already demonstrated that our dataset generation pipeline could be extended to other actions. In fact, we provided examples of two additional actions, `turn` and `jump`, in Fig.3 in the Supp. Mat **in the initial submission**. Here, we further demonstrate the pipeline's capability with both interactive actions (such as `open`, `place`, `knock`) and non-interactive actions (such as `dance`). Please visit the [anonymous website](https://dsdbhj.github.io/humanise_material/index.html#dataset) for visualizations.

### More fine-grained interactions

Given existing methods, generating high-quality and large-scale fine-grained HSIs such as hand-object interactions in dynamic 3D scenes is much more challenging. How to collect such a dataset is still an open problem. It could be an interesting direction for future work.

## 2. Clarification of motion duration

### Design principle

We follow previous works to generate human motion ranging from 30 to 120 frames (30 FPS). For example, the most related work [6] generates human motions that last 2 seconds in 3D scenes, ACTOR [5] generates human motions that are less than 120 frames, and [2] considers predicting the human motion in the future 3 seconds with given histories.

### Synthesize longer motions

On the one hand, our dataset generation pipeline can synthesize longer motions. We provide examples on the [anonymous website](https://dsdbhj.github.io/humanise_material/index.html#longer), including longer `walk` and `dance` that last more than 8 seconds. On the other hand, longer human motions can be seen as the composition of atomic action sequences. With our work, long-duration human motion generation and composition are promising in the future.

## 3. Clarification of language descriptions

The language descriptions we provide in HUMANISE act as the first step to providing sufficient information about the action and scene semantics for goal-oriented human motion generation. Although the descriptions are generated using templates, they are diverse in terms of the referential utterances [3] and the involved spatial relations; see main paper Sec. 3.2. Note that the 3D object grounding accuracy on the Sr3D dataset [3] is less than $40\%$, meaning that such a templated description is still challenging for models to locate the target objects and generate motions. We further showcase the generalization ability of our model on natural descriptions from human annotators in (1) the test split of HUMANISE, and (2) the new dataset Replica [12]. Some sample results are shown in the [anonymous website](https://dsdbhj.github.io/humanise_material/index.html#generalize).
In our future work, we will further explore the generalization ability to use natural language description and other 3D scene datasets.

## 4. Steps toward natural Human-Scene Interaction (HSI) understanding and generation

To solve the challenges in HSI understanding and generate motions precisely with instructions, ideally, we hope to collect a large-scale dataset where long and diverse motions interact with dynamic 3D scenes and are annotated with natural language descriptions. However, this is still challenging with current 3D capturing techniques and high costs. Our work sufficiently bridges existing and future works by proposing HUMANISE with large-scale language-conditioned motions and enabling a new language-conditioned generation task. We will release our dataset, extendable dataset generation pipeline, and models to facilitate the HSI research.

## Updates to the initial submission

We have revised the typos in the main paper and the Supp. Mat. as the reviewers suggest, shown in red.

**Main paper:**

- Fixing typos in Table 1, L2, L87, L114,
- L124: Adding how to sample the interacting object by the action label
- L248: Adding the scene number in the train/test split.

**Supp. Mat:**

- L16-L20: Adding motion diversity statistics when generating HUMANISE.

---

> ### Author Response · Authors · 2022-08-02
> **Bibliography**
>
> [1] Mohamed Hassan, Vasileios Choutas, Dimitrios Tzionas, and Michael J Black. Resolving 3d human pose ambiguities with 3d scene constraints. In _International Conference on Computer Vision (ICCV)_, 2019.
> [2] Zhe Cao, Hang Gao, Karttikeya Mangalam, Qi-Zhi Cai, Minh Vo, and Jitendra Malik. Long-term human motion prediction with scene context. In _European Conference on Computer Vision (ECCV)_, 2020.
> [3] Panos Achlioptas, Ahmed Abdelreheem, Fei Xia, Mohamed Elhoseiny, and Leonidas Guibas. Referit3d: Neural listeners for fine-grained 3d object identification in real-world scenes. In _European Conference on Computer Vision (ECCV)_, 2020.
> [4] Mohamed Hassan, Duygu Ceylan, Ruben Villegas, Jun Saito, Jimei Yang, Yi Zhou, and Michael J Black. Stochastic scene-aware motion prediction. In _International Conference on Computer Vision (ICCV)_, 2021.
> [5] Mathis Petrovich, Michael J Black, and Gül Varol. Action-conditioned 3d human motion synthesis with transformer vae. In _International Conference on Computer Vision (ICCV)_, 2021.
> [6] Jiashun Wang, Huazhe Xu, Jingwei Xu, Sifei Liu, and Xiaolong Wang. Synthesizing long-term 3d human motion and interaction in 3d scenes. In _Conference on Computer Vision and Pattern Recognition (CVPR)_, 2021.
> [7] Abhinanda R Punnakkal, Arjun Chandrasekaran, Nikos Athanasiou, Alejandra Quiros-Ramirez, and Michael J Black. Babel: Bodies, action and behavior with english labels. In _Conference on Computer Vision and Pattern Recognition (CVPR)_, 2021.
> [8] Yan Zhang, Mohamed Hassan, Heiko Neumann, Michael J Black, and Siyu Tang. Generating 3d people in scenes without people. In _Conference on Computer Vision and Pattern Recognition (CVPR)_, 2020b.
> [9] Mohamed Hassan, Partha Ghosh, Joachim Tesch, Dimitrios Tzionas, and Michael J Black. Populating 3d scenes by learning human-scene interaction. In _Conference on Computer Vision and Pattern Recognition (CVPR)_, 2021b.
> [10] Kaifeng Zhao, Shaofei Wang, Yan Zhang, Thabo Beeler and Siyu Tang. Compositional Human-Scene Interaction Synthesis with Semantic Control. In _European Conference on Computer Vision (ECCV)_, 2022.
> [11] Naureen Mahmood, Nima Ghorbani, Nikolaus F Troje, Gerard Pons-Moll, and Michael J Black. Amass: Archive of motion capture as surface shapes. In _International Conference on Computer Vision (ICCV)_, 2019.
> [12] Julian Straub, Thomas Whelan, Lingni Ma, Yufan Chen, Erik Wijmans, Simon Green, Jakob J Engel, Raul Mur-Artal, Carl Ren, Shobhit Verma, et al. The replica dataset: A digital replica of indoor spaces. _arXiv preprint arXiv:1906.05797_, 2019.

---

### Meta-Review · Area_Chair_j7oy · 2022-08-24

**Recommendation:** Accept
**Confidence:** Certain

**Metareview:**

Paper was reviewed by four reviewers and received: 1 x Borderline Accept, 1 x Borderline Reject, 1 x Weak Accept and 1 x Accept. The general sentiment of reviewers was positive. Main identified concerns were with lack of diversity in the dataset and potential realism issues arising from construction of the dataset (placing pre-recorded motions into different scenes). Some of these concerns have been somewhat alleviated through the rebuttal. That said, [bBY2] remained concerned with the realism of interactions. AC agrees with [vzPA] that collecting "natural" real-world data for this problem would be difficult and laborious and that the proposed dataset can serve as a good bridge towards this ultimate goal and could be a stepping stone for future research. Therefore the decision is to Accept the paper.

**Award:**

No

---

### Decision · Program_Chairs · 2022-09-14

Accept